# Using repeated home-based HIV testing services to reach and diagnose HIV infection among persons who have never tested for HIV, Chókwè health demographic surveillance system, Chókwè district, Mozambique, 2014–2017

Carol Lin[1]*, Isabelle Casavant[2], Alicia Jaramillo[3], Timothy Green[4]

1 Division of Global HIV and TB, Center for Global Health, U.S. Centers for Disease Control and Prevention, Atlanta, Georgia, United States of America, 2 U.S. Centers for Disease Control and Prevention, Maputo, Mozambique, 3 Jhpiego Corporation, Maputo, Mozambique, 4 Division of HIV/AIDS Prevention, National Center for HIV/AIDS, Viral Hepatitis, STD and TB Prevention, U.S. Centers for Disease Control and Prevention, Atlanta, Georgia, United States of America

* clin@cdc.gov

## Abstract

### Background

HIV prevalence in Mozambique (12.6%) is one of the highest in the world, yet ~40% of people living with HIV (PLHIV) do not know their HIV status. Strategies to increase HIV testing uptake and diagnosis among PLHIV are urgently needed. Home-based HIV testing services (HBHTS) have been evaluated primarily as a 1-time campaign strategy. Little is known about the potential of repeating HBHTS to diagnose HIV infection among persons who have never been tested (NTs), nor about factors/reasons associated with never testing in a generalized epidemic setting.

### Methods

During 2014–2017, counselors visited all households annually in the Chókwè Health and Demographic Surveillance System (CHDSS) and offered HBHTS. Cross-sectional surveys were administered to randomly selected 10% or 20% samples of CHDSS households with participants aged 15–59 years before HBHTS were conducted during the visit. Descriptive statistics and logistic regression were used to assess the proportion of NTs, factors/reasons associated with never having been tested, HBHTS acceptance, and HIV-positive diagnosis among NTs.

### Results

The proportion of NTs decreased from 25% (95% confidence interval [CI]:23%–26%) during 2014 to 12% (95% CI:11% –13%), 7% (95% CI:6%–8%), and 7% (95% CI:6%–8%) during

**Data Availability Statement:** All relevant data are within the manuscript and its Supporting Information files.

**Funding:** The project has been supported by the President's Emergency Plan for AIDS Relief (PEPFAR) through the Centers for Disease Control and Prevention (CDC) under the terms of [CoAg#GH00080]. The findings and conclusions in this report are those of the authors and do not necessarily represent the official position of the funding agencies.

**Competing interests:** The authors have declared that no competing interests exist.

2015, 2016, and 2017, respectively. Adolescent boys and girls and adult men were more likely than adult women to be NTs. In each of the four years, the majority of NTs (87%–90%) accepted HBHTS. HIV-positive yield among NTs subsequently accepting HBHTS was highest (13%, 95% CI:10%–15%) during 2014 and gradually reduced to 11% (95% CI:8%–15%), 9% (95% CI:6%–12%), and 2% (95% CI:0%–4%) during 2015, 2016, and 2017, respectively.

## Conclusions

Repeated HBHTS was helpful in increasing HIV testing coverage and identifying PLHIV in Chókwè. In high HIV-prevalence settings with low testing coverage, repeated HBHTS can be considered to increase HIV testing uptake and diagnosis among NTs.

## Introduction

HIV remains one of the world's most serious public health challenges. Approximately, 38 million people worldwide are living with HIV/AIDS and 1.7 million people became newly infected with HIV in 2019 [1]. Mathematical models and observational studies conclude that HIV test-and-treat strategies hold great potential for reducing HIV transmission, morbidity, and mortality in generalized epidemic settings [2–5]. Recent evidence has also established that persons who take antiretroviral therapy (ART) daily as prescribed and achieve and maintain an undetectable viral load have effectively no risk for transmitting the virus to an HIV-negative sex partner [6–8]. The success of HIV test-and-treat strategies depends on HIV testing uptake, linkage to care, and adherence to ART among HIV-positive persons. However, HIV testing uptake remains low: a quarter of persons living with HIV (PLHIV) worldwide remain unaware of their infection status and present at clinics at a late disease stage [9]. Identifying effective HIV testing strategies to increase uptake, especially among those persons who have never been tested (NTs), and assessing factors and reasons associated with never testing remain crucial.

With an estimated 2,200,000 people living with HIV, adult (ages 15–49) HIV prevalence in Mozambique (12.6%) in 2018 was one of the highest in the world [10, 11]. In 2018, an estimated 150,000 new HIV infections were identified and 5,000 AIDS-related deaths occurred [11]. Implementing effective HIV testing and linkage to ART services are key strategies employed by Mozambique's government for preventing HIV transmission [12]. Although uptake of HIV testing has increased, the overall testing coverage in Mozambique remains low. Approximately 40% of PLHIV have never been tested and do not know their HIV status [11]. Strategies for increasing HIV testing uptake, especially, among people who have never been tested for HIV, and diagnosis of HIV infection are urgently needed.

Facility-based HIV testing services (FBHTS), including voluntary, provider-initiated HIV testing services (HTS), have not been sufficient to meet the Joint United Nations Programme on HIV/AIDS (UNAIDS) 90-90-90 target (By 2020, 90% of all people living with HIV know their HIV status; 90% of all people with diagnosed HIV infection receive sustained antiretroviral therapy; 90% of all people receiving antiretroviral therapy have viral suppression) and the associated goals for bringing the global HIV epidemic under control by 2020 in sub-Saharan Africa [13, 14]. Community-based HTS, including home-based HTS (HBHTS), mobile-based HTS (MBHTS), and index testing, are additional strategies to increase HIV testing uptake [15–20]. The HBHTS strategy consists of offering HIV testing and counseling to individuals encountered at their home, for all homes in a defined geographic area. HBHTS can overcome

HIV-testing barriers, including lack of knowledge and distance to testing sites, long wait times, transportation costs, lost wages, costs associated with childcare, and concerns about confidentiality and stigma [17, 21, 22]. HBHTS also allows couples and families to be counseled together about HIV testing, HIV risk reduction, and ART [17]. A meta-analysis of 28 studies concluded that community testing achieves higher testing uptake and helps to identify HIV-positive persons at an earlier stage of their infection (i.e., at higher $CD4^+$ counts) than FBHTS, but the proportion of new HIV-positive diagnoses (yield) by FBHTS is higher [17]. Similarly, a randomized controlled trial in Lesotho comparing HBHTS with MBHTS demonstrated that HBHTS can achieve higher testing uptake, but MBHTS detects a higher proportion of new HIV infections [23]. Recently, a study conducted in South Africa compared index testing to other community testing modalities (mobile, homebased or workplace) and concluded that index testing identified higher proportions of HIV-positive persons than other modalities overall but the proportions of HIV uptake and positive diagnosis by index testing among persons aged 25–49 years were lower [8]. Another meta-analysis of 21 studies during 2002–2012 in 5 African countries concluded that HBHTS can substantially increase previously undiagnosed persons' awareness of their HIV status [24]. These HBHTS evaluation studies have been focused on a one-time campaign approach (i.e., a single period during which HTS teams travel door-to-door and offer HTS).

Helleringer et al. (2013) [22] evaluated repeated HBHTS through 2 HBHTS campaigns in Likoma, Malawi, but the evaluation was limited to overall acceptance of HBHTS, HIV prevalence, and associated costs. Little is known about the potential of annually repeating HBHTS to reach and diagnose HIV infection among NTs, nor about factors and reasons associated with never testing. This information is needed to assess the value of the HBHTS strategy in helping countries to achieve >90% awareness of status among HIV-infected persons. Without testing, undiagnosed HIV-seropositive individuals will not receive the treatment they need to slow disease progression and prevent transmission. Additionally, understanding the different factors or reasons associated with never testing may be helpful in developing and identifying strategies for increasing HIV testing uptake. For this paper, we used HIV prevention survey (HPS) and HBHTS data collected annually during 2014–2017 (4 different rounds) through the Chókwè Health Demographic Surveillance System (CHDSS) in Chókwè District, Mozambique, to assess (a) the proportion of NTs aged 15–59 before and after HBHTS were implemented, (b) factors or reasons associated with never testing, and (c) HBHTS acceptance and positive diagnosis results (yield) among NTs.

## Methods

### Setting and study design

Chókwè District is located in Gaza Province and has the highest adult (15–49 yrs old) HIV prevalence (25%) in Mozambique [10, 25]. A Health Demographic Surveillance System (HDSS) was first established by Chókwè Health Research and Training Center (CITSC) in 2010. CHDSS covers Chókwè city and several neighborhood villages which, together, include approximately 100,000 of the total Chókwè district population of 183,000, and approximately 58,000 residents aged 15–59 years. During 4 separate rounds between 2014–2017, HTS counselors visited all CDHSS households and offered HBHTS to household members who had not previously tested HIV- positive. Households with at least one eligible member who had not been offered HTS were revisited at least one more time. During these rounds, HPS was offered to household members aged 15–59 years in a randomly selected 10% (Rounds 1–2, 2014–2015) or 20% (Rounds 3–4, 2016–2017) sample of CHDSS households.

For participants aged 15–17, parental informed written consent was obtained. For all eligible participants with informed written consent to both the HPS and HBHTS, a 20–30 minute HPS questionnaire was administered by trained interviewers in Portuguese or Shangana before HBHTS was conducted. The study protocol was reviewed and approved by the CHDSS community advisory board, National AIDS Control Program of the Mozambique Ministry of Health, and Mozambique National Health Bioethics Committee as research and was determined to be non-engaged research by the CDC Center for Global Health.

## HIV testing and counseling

Rapid HIV testing and confidential pre- and post-test counseling were provided by trained counselors at CDHSS participants' homes according to Mozambique's national guidelines. HIV-positive participants were provided additional HIV counseling (e.g., referrals, linkage to care, and information about the benefits of early treatment, adherence, disclosure of status, partner or family HIV testing, and condom use). Counselors conducted up to 5 follow-up home visits to encourage HIV-positive participants to enroll in and adhere to HIV care. HIV-negative and HIV-indeterminate participants were provided risk-reduction counseling, including recommendations for periodic HIV testing and behavioral prevention strategies/services; uncircumcised men were referred to voluntary medical male circumcision services. Pregnant women were referred for antenatal care when needed.

## Outcomes and other HIV testing-related variables

The primary outcome variable was never having been tested for HIV. All persons surveyed were asked, "Have you ever been tested for HIV?" Additional outcome variables included acceptance of HBHTS among NTs, and HIV-positive yield among NTs who tested for HIV after their survey interview. For participants who responded "Yes" to ever testing for HIV, information about the location of their most recent HIV test was collected. For participants who responded "No" to ever having been tested for HIV, the reasons for not having been tested and their intention to test for HIV during the next 12 months were assessed.

## Demographic, behavioral, and psychological variables

The HPS questionnaire included standard measures on demographics (sex, age, and marital status), drug use (marijuana or other during the prior 3 months), experience of physical or sexual violence (during the prior 12 months), and sexual history. Sexual history-related measures included number of sex partners during the prior 12 months, status of the most recent sexual partner (i.e., spouse, casual or exchange sex partner), if a condom was used during most recent sexual act, and if the participant asked about the sex partner's HIV and sexually transmitted infection status. In addition, a series of questions related to comprehensive HIV knowledge, beliefs about ART, and HIV/AIDS stigma were also asked. Specific questions (multiple choice questions) or items included in these three composite measures are available in the S1 File.

## Statistical analysis

The analysis was based on all HPS participants. Around 65%, 72%, 69% and 66% of the eligible participants were reached during the 4 survey rounds. Among those reached, the survey refusal rates were 15%, 15%, 15% and 21%. The observed data were analyzed using SAS® survey procedures (PROC SUVEYFREQ and SURVEYLOGISTIC; version 9.3, SAS Institute, Inc., Cary, North Carolina, USA) that account for correlations among participants within a household [26]. Summary statistics of demographic and behavioral characteristics, knowledge about

HIV, beliefs about ART, and stigma scores of all survey respondents and NTs were calculated. Additionally, the proportions of respondents tested for HIV by testing location, and whether or not HIV testing was discussed with partners, were calculated. Similarly, the proportions of NTs by reasons for never having been tested, intention to have an HIV test during the next 12 months, HBHTS acceptance, and HIV test positivity among NTs subsequently accepting HBHTS were calculated.

To assess the changes among NTs, the number and proportion of NTs in each round, including 95% Wald confidence intervals (CIs), were calculated. To identify the factors associated with never having been tested, bivariate and multivariable analyses were conducted for each survey round using the annual data. Logistic regression analyses were used to identify factors associated with never having been tested. Factors with a P-value<0.1 in the bivariate analyses were included in the initial multivariable models. Backward elimination was used to remove, one at a time, the factor with the highest P-value in the multivariable model until only factors with a P-value<0.05 remained. Two-way interaction terms between the remaining factors were then evaluated and the final model included all terms with a P-value less than 0.05.

To assess the magnitude of HBHTS acceptance among NTs and HIV-positive diagnoses among NTs accepting HBHTS, the proportions of NTs accepting HBHTS and with HIV-positive test results, including 95% Wald or Wilson CIs, were calculated for each round. To assess the relationship between acceptance of HBHTS among NTs and reasons for not testing, we combined the data over survey rounds (2014–2017). Proportions of NTs accepting HBHTS and diagnosed as HIV-positive by the statistically significant factors identified (i.e., sex and age) by logistic regression and by reported reasons for never having been tested were estimated.

To test the robustness of the observed findings, we then conducted a sensitivity analysis [27]. We re-analyzed the data using a weighted approach with a survey weight calculated by age, gender and region (urban or rural).

## Results

Survey participants' demographic, behavioral, and psychological characteristics were similar in all survey rounds (Table 1). The proportions of survey respondents who were female ranged from 63% to 73%; 58%–62% were aged ≥25 years; 49%–56% were married; and 85%–90% had ever had sex. The proportions of respondents with >1 sex partner, knowing persons who had died with AIDS, and engaging in sexual risk behaviors (i.e., having unprotected sex or never asking partners about HIV status) decreased over time. The median scores for HIV knowledge, beliefs about ART, and stigma were 7–8 (out of 9), 5 (out of 6), and 8–10 (out of 16), respectively.

### Proportion of NTs and HIV testing-associated characteristics

The proportion of NTs decreased over time, from 25% (95% CI: 23%–26%) during 2014 to 12% (95% CI: 11%–13%), 7% (95% CI: 6%–8%), and 7% (95% CI: 6%–8%) during subsequent years. During the first round before HBHTS was first implemented, the most frequently reported location for the most recent HIV test was hospitals in Chókwè (45%), followed by home in Chókwè (13%), other location in Chókwè (10%), and at work in Chókwè (2%). After HBHTS was implemented, home in Chókwè became the most frequently reported location for the most recent HIV test, increasing from 13% during 2014 to 50%, 50%, and 49% during the 2015–2017 rounds, respectively. Hospitals in Chókwè became the second most frequently reported location (41%–43%) during the 2nd to 4th rounds.

**Table 1. Sample characteristics and proportions of participants never having been tested for HIV, by survey round (2014–2017).**

| | 2014 | | | 2015 | | | 2016 | | | 2017 | | |
|---|---|---|---|---|---|---|---|---|---|---|---|---|
| | Total No. | Never Tested Perc. | | Total No. | Never Tested Perc. | | Total No. | Never Tested Perc. | | Total No. | Never Tested Perc. | |
| | N | % | | N | % | | N | % | | N | % | |
| | UW/W | UW | W | UW/W | UW | W | UW/W | UW | W | UW/W | UW | W |
| | 3024/3027 | 25 | 24 | 3151/3148 | 12 | 12 | 5061/5049 | 7 | 8 | 4415/4415 | 7 | 7 |
| **Sex** | | | | | | | | | | | | |
| Male | 1115/1164 | 37 | 35 | 887/1211 | 18 | 17 | 1381/1932 | 12 | 13 | 1326/1695 | 10 | 9 |
| Female | 1909/1863 | 18 | 17 | 2264/1937 | 9 | 9 | 3680/3117 | 6 | 6 | 3089/2720 | 6 | 6 |
| **Age** | | | | | | | | | | | | |
| <18 | 434/431 | 49 | 48 | 508/543 | 33 | 30 | 848/905 | 25 | 26 | 816/772 | 30 | 29 |
| 18–24 | 758/806 | 26 | 24 | 728/744 | 9 | 10 | 1090/1171 | 5 | 5 | 1021/1036 | 2 | 2 |
| $\geq 25$ | 1832/1790 | 18 | 18 | 1915/1860 | 7 | 8 | 3123/2973 | 3 | 4 | 2578/2607 | 2 | 2 |
| **Relationship** | | | | | | | | | | | | |
| Married/Marital union | 1699/1690 | 18 | 18 | 1748/1668 | 7 | 7 | 2625/2506 | 4 | 5 | 2178/2145 | 13 | 12 |
| Other | 1324/1336 | 33 | 32 | 1402/1478 | 18 | 18 | 2429/2537 | 11 | 11 | 2236/2270 | 17 | 22 |
| **Knowing people died with AIDS** | | | | | | | | | | | | |
| No | 1728/1716 | 26 | 25 | 2167/2210 | 11 | 12 | 3422/3395 | 6 | 7 | 3052/3077 | 6 | 6 |
| Yes | 1035/1091 | 17 | 17 | 782/765 | 7 | 8 | 1099/1131 | 4 | 6 | 868/803 | 3 | 3 |
| **Ever have sex** | | | | | | | | | | | | |
| No | 302/293 | 50 | 50 | 368/395 | 36 | 34 | 615/662 | 26 | 25 | 679/672 | 31 | 28 |
| Yes | 2711/2721 | 22 | 21 | 2778/2747 | 8 | 9 | 4427/4372 | 5 | 6 | 3732/3740 | 3 | 2 |
| **Number of sex partners in the past 12 months** | | | | | | | | | | | | |
| ≤1 | 2211/2177 | 22 | 22 | 2660/2523 | 11 | 12 | 3969/3736 | 7 | 8 | 3811/3678 | 7 | 7 |
| >2 | 794/828 | 31 | 29 | 449/574 | 15 | 14 | 766/993 | 8 | 9 | 352/440 | 8 | 7 |
| **Having casual sex or exchange Partner** | | | | | | | | | | | | |
| No | 2112/2103 | 22 | 22 | 2302/2287 | 11 | 12 | 3663/3689 | 8 | 9 | 3193/3208 | 8 | 8 |
| Yes | 736/752 | 30 | 29 | 582/649 | 15 | 15 | 972/1040 | 8 | 8 | 891/941 | 6 | 5 |
| **Having unprotected sex (no condom) with last sex partner** | | | | | | | | | | | | |
| No | 922/978 | 28 | 27 | 964/1078 | 19 | 18 | 1654/1853 | 13 | 13 | 1735/1855 | 14 | 12 |
| Yes | 1877/1825 | 23 | 22 | 1893/1835 | 9 | 9 | 2914/2815 | 5 | 6 | 2308/2255 | 25 | 30 |
| **Never asked partner about HIV status when have sex** | | | | | | | | | | | | |
| No | 1625/1672 | 19 | 18 | 1678/1752 | 11 | 12 | 2906/2977 | 8 | 8 | 2722/2748 | 9 | 8 |
| Yes | 1199/1160 | 32 | 32 | 1169/1149 | 13 | 14 | 1648/1678 | 8 | 10 | 1307/1353 | 6 | 6 |
| **Drug use in the last 3 months** | | | | | | | | | | | | |
| No | 2897/2889 | 24 | 23 | 3092/3064 | 11 | 12 | 4983/4952 | 7 | 8 | 4334/4308 | 7 | 7 |
| Yes | 127/138 | 39 | 36 | 59/84 | 20 | 20 | 78/97 | 18 | 16 | 81/108 | 6 | 10 |
| **Having STI in the past 12 months** | | | | | | | | | | | | |
| No | 2185/2174 | 27 | 27 | 2497/2523 | 13 | 13 | 4226/4261 | 8 | 9 | 3822/3811 | 8 | 8 |
| Yes | 839/853 | 17 | 16 | 654/625 | 6 | 7 | 835/787 | 4 | 6 | 593/605 | 1 | 1 |
| **Partner violence in the past 12 months** | | | | | | | | | | | | |
| No | 2683/2652 | 25 | 24 | 2948/2950 | 12 | 13 | 4777/4807 | 8 | 9 | 4190/4170 | 7 | 7 |
| Yes | 315/348 | 20 | 20 | 192/186 | 6 | 6 | 126/142 | 6 | 6 | 148/167 | 1 | 1 |
| | Total | Never tested | | Total | Never tested | | Total | Never tested | | Total | Never Tested | |
| | Median (Q1, Q3) | Median (Q1, Q3) | | Median (Q1, Q3) | Median (Q1, Q3) | | Median (Q1, Q3) | Median (Q1, Q3) | | Median (Q1, Q3) | Median (Q1, Q3) | |

(*Continued*)

**Table 1.** (Continued)

|  | 2014 | | | 2015 | | | 2016 | | | 2017 | | |
|  | Total No. | Never Tested Perc. | | Total No. | Never Tested Perc. | | Total No. | Never Tested Perc. | | Total No. | Never Tested Perc. | |
|  | N | % | | N | % | | N | % | | N | % | |
|  | UW/W | UW | W | UW/W | UW | W | UW/W | UW | W | UW/W | UW | W |
| **Knowledge about HIV Score** (max: 9) | 7 (6,8)/7(6,8) | 7(5,8) | 7(5,8) | 7(6,8)/7(6,8) | 6(4,8) | 6(4.8) | 8(6,9)/7(6,9) | 6 (4,8) | 6 (5,8) | 7(6,8)/7(6,8) | 6 (4,7) | 6 (4,7) |
| **Belief about ART Score** (max: 6) | 5(3,5)/5(4,5) | 4(2,5) | 4(2,5) | 5(4,6)/5(4,6) | 5(2,6) | 5(2,5) | 5(4,6)/5(4,6) | 5 (0,5) | 5 (0,5) | 5(3,5)/4(3,5) | 3 (0,5) | 3 (0,5) |
| **Stigma Score** (max:16) | 10(8,10)/10 (8,10) | 9 (8,10) | 9 (8,10) | 8(8,10)/8 (8,10) | 8 (8,10) | 8 (8,10) | 8(8,8)/8(8,8) | 8 (8,8) | 8 (8,8) | 8(8,8)/8(8,8) | 8 (8,8) | 8 (8,8) |

UW: unweighted, observed data.

W: weighted.

## Factors and reasons associated with never having been tested

Factors associated with NT, after adjusting for all the other variables in the model, were similar for all 4 survey rounds (Table 2). During survey rounds 1–3, adolescent (ages 15–17 years) boys and girls and adult (ages ≥18) men were more likely than adult women to be NTs ($P <$ .001). Adolescent girls had 4.91 (95% CI: 3.16–7.63), 6.23 (4.02–9.64), and 16.17 (6.86–38.16) times the adjusted odds of never having been tested, compared with adult women in 2014, 2015, and 2016, respectively. Adult men had 3.64 (95% CI: 2.88–4.61), 3.62 (2.52–5.20), and 3.29 (2.17–5.00) times the adjusted odds of never having been tested, compared with adult women in 2014, 2015, and 2016, respectively. Age (but not sex) remained a statistically significant factor in 2017, with adolescents having 9.50 (95% CI: 5.85–15.41) times the adjusted odds of never having been tested, compared with adults (ages≥25). Participants who had ever had sex, who had higher HIV knowledge scores, or who had ever asked partners about HIV status were less likely to be NTs for each of the 4 years.

The most frequently reported reasons for never having been tested were similar across survey rounds (Table 3). Frequently reported reasons for never have been tested during 2014, 2015, 2016, and 2017, respectively, were limited access or time (26%, 35%, 25%, and 27%), indifference (i.e., does not know, want to know, care, or think about whether or not they are infected) (19%, 21%, 20%, and 17%), low perceived risk for HIV infection (33%, 16%, 20%, and 21%), lack of being offered testing by a healthcare provider (12%, 12%, 17% and 22%), and fear of needles, blood or testing-HIV positive (9%, 7%, 16%, and 12%).

## HIV testing acceptance among NTs and HIV-positivity diagnosis among NTs accepting HBHTS

The acceptance rate of HBHTS among NTs was high throughout all 4 years (Table 4). After the survey was administered, 90% (95% CI: 87%–92%), 87% (95% CI: 83%–90%), 88% (95% CI: 83%–91%), and 88% (95% CI: 83%–91%) of NTs accepted HBHTS during 2014–2017. The yield of new HIV-positive diagnoses among NTs subsequently accepting HBHTS, was 13% (95% CI: 10%–15%) during 2014 and gradually reduced to 11% (95% CI: 8%–15%), 9% (95% CI: 6%–12%), and 2% (95% CI: 0%–4%) during 2015, 2016, and 2017. Among NTs aged ≥25 years (98% of whom had ever had sex) offered HBHTS, the proportion of HIV-positive diagnosis was ~20% during the first 3 years but decreased to 5% during 2017.

**Table 2. Demographic and behavioral factors associated with never tested for HIV before.**

| | 2014 | | 2015 | | 2016 | | 2017 | |
|---|---|---|---|---|---|---|---|---|
| | Crude OR (95% CI) | Adjusted OR (95% CI) | Crude OR (95% CI) | Adjusted OR (95% CI) | Crude OR (95% CI) | Adjusted OR (95% CI) | Crude OR (95% CI) | Adjusted OR (95% CI) |
| **Sex** | | | | | | | | |
| Male | 2.71 (2.27, 3.33) | | 2.35 (1.88, 3.00) | | 2.21 (1.78, 2.74) | | 1.81 (1.43, 2.28) | |
| Female | 1 | | 1 | | 1 | | 1 | |
| **Age** | | | | | | | | |
| <18 | 4.37 (3.45, 5.52) | | 6.80 (5.25, 8.82) | | 10.0 (7.80, 13.00) | | 27.71 (19.33, 39.73) | 9.50 (5.85,15.41) |
| 18–24 | 1.56 (1.27, 1.92) | | 1.33 (0.97, 1.84) | | 1.53 (1.09, 2.15) | | 1.59 (0.96, 2.65) | 1.23 (0.69, 2.21) |
| ≥ 25 | 1 | | 1 | | 1 | | 1 | 1 |
| **Relationship** | | | | | | | | |
| Other | 2.16 (1.81, 2.57) | | 2.98 (2.34, 3.78) | | 3.17 (2.49, 4.02) | | 8.62 (6.60, 12.39) | |
| **Married/martial union** | 1 | | 1 | | 1 | | 1 | |
| **Knowing people died with AIDS** | | | | | | | | |
| No | 1.60 (1.32, 1.95) | | 1.85 (1.32, 2.57) | | 1.54 (1.11, 2.15) | | 2.19 (1.40, 3.42) | |
| Yes | 1 | | 1 | | 1 | | 1 | |
| **Ever have sex** | | | | | | | | |
| No | 3.59 (2.79, 4.64) | 6.65 (4.10, 10.81) | 6.12 (4.73, 7.91) | 6.34 (4.02, 10.14) | 7.01 (5.56, 8.84) | | 15.26 (11.74, 19.83) | 5.85 (3.45, 9.93) |
| Yes | 1 | 1 | 1 | 1 | 1 | | 1 | 1 |
| **Having casual sex or exchange partner (last person had sex with)** | | | | | | | | |
| No | 0.66 (0.54, 0.80) | 0.68 (0.52, 0.89) | 0.74 (0.57, 0.97) | 0.57 (0.40, 0.82) | 1.06 (0.81, 1.39) | | 1.42 (1.03, 1.94) | |
| Yes | 1 | 1 | 1 | 1 | 1 | | 1 | |
| **Having unprotected sex (no condom) with last sex partner** | | | | | | | | |
| No | 1.28 (1.06, 1.54) | 0.52 (0.38,0.69) | 2.43 (1.93, 3.07) | | 2.78 (2.24, 3.47) | | 6.13 (4.51, 8.34) | |
| Yes | 1 | | 1 | | 1 | | | |
| **Never asked partner about HIV status when have sex** | | | | | | | | |
| No | 0.49 (0.40, 0.57) | 0.32 (0.25, 0.41) | 0.85 (0.67, 1.07) | 0.34 (0.24, 0.49) | 0.90 (0.72, 1.12) | 0.39 (0.27, 0.58) | 1.61 (1.23, 2.13) | 0.33 (0.20, 0.53) |
| Yes | 1 | 1 | 1 | 1 | 1 | 1 | 1 | 1 |
| **Having STI in the past 12 month** | | | | | | | | |
| No | 1.82 (1.49, 2.22) | | 2.15 (1.55, 3.00) | | 2.13 (1.48, 3.06) | | 5.60 (2.87, 10.92) | |
| Yes | 1 | | 1 | | 1 | | 1 | |
| **Drug use** | | | | | | | | |
| No | 0.48 (0.34, 0.70) | | 0.50 (0.27, 0.95) | 0.49 (0.23, 1.06) | 0.36 (0.20, 0.65) | | 1.16 (0.47, 2.90) | |
| Yes | 1 | | 1 | 1 | 1 | | 1 | |
| **Score: Knowledge about HIV** | 0.84 (0.80, 0.89) | 0.88 (0.83, 0.93) | 0.80 (0.74, 0.85) | 0.88 (0.82, 0.95) | 0.74 (0.71, 0.78) | 0.88 (0.82, 0.93) | 0.73 (0.69, 0.78) | 0.93 (0.86, 1.00) |
| **Score: believe of ARV** | 0.89 (0. 85, 0.93) | | 0.88 (0.83, 0.93) | | 0.83 (0.79, 0.88) | | 0.81 (0.76, 0.86) | |

(*Continued*)

**Table 2.** (Continued)

| | 2014 | | 2015 | | 2016 | | 2017 | |
|---|---|---|---|---|---|---|---|---|
| | Crude OR (95% CI) | Adjusted OR (95% CI) | Crude OR (95% CI) | Adjusted OR (95% CI) | Crude OR (95% CI) | Adjusted OR (95% CI) | Crude OR (95% CI) | Adjusted OR (95% CI) |
| **Score: stigma** | 1.05 (0.99, 1.09) | | 1.11 (1.02, 1.18) | | 1.11 (1.06, 1.17) | 1.10 (1.03, 1.19) | 1.08 (0.98, 1.20) | |
| **Interaction: age and gender** | | | | | | | | |
| Age<18 and male | | 3.65 (2.44, 5.45) | | 2.82 (1.69, 4.70) | | 13.40 (5.58, 32.02) | | |
| Age<18 and female | | 4.91 (3.16, 7.63) | | 6.23 (4.02, 9.64) | | 16.17 (6.86, 38.16) | | |
| Age> = 18 and male | | 3.64 (2.88, 4.61) | | 3.62 (2.52, 5.20) | | 3.29 (2.17, 5.00) | | |
| Age> = 18 and female | | 1 | | 1 | | 1 | | |
| **Interaction: age and ever have sex** | | | | | | | | |
| Age<18 and never have sex | | | | | | 13.46 (8.90, 20.37) | | |
| Age<18 and have sex | | | | | | 3.21 (2.07, 4.98) | | |
| Age> = 18 and never have sex | | | | | | 0.66 (0.15, 2.93) | | |
| **Age> = 18 and have sex** | | | | | | 1 | | |

No evidence of lack of fit for all models.

When the data for all 4 survey rounds were combined, HBHTS was accepted by >80% of NTs in each sex and age group, and among NTs reporting each reason for never testing previously, except for fear of needles, blood, or testing HIV-positive (Table 4). Of 498 NTs who reported limited access or time as reasons for never testing, 470 (94%) accepted HBHTS, of whom 11% tested HIV-positive. Of 437 and 340 NTs who reported low perceived risk or being indifferent as reasons for never testing previously, 389 (89%) and 291 (86%) accepted HBHTS, of whom 7% and 10% tested HIV-positive, respectively. Of 185 NTs who reported being afraid of needles, blood, or testing HIV-positive as a reason for never previously testing, 131 (71%) accepted HBHTS, of whom, 9% tested HIV-positive.

The sensitivity analysis results using the weighted approach are given in Table 1 and S1–S3 Tables. The weighted and unweighted estimates were similar. In particular, the estimated proportions of participants never having been tested by different characteristics and the proportions of HBHTS acceptance and HBHTS positive among persons who had never been tested obtained using the two approaches were very close. In addition, the factors associated with never having been tested were the same with the two approaches.

## Discussion

During 2014–2017, we used 4 rounds of HPS and HBHTS data to investigate the potential of repeating HBHTS annually for reaching NTs, and to determine factors or reasons associated with never having been tested previously, acceptance of HBHTS, and yield of new HIV diagnoses among NTs who participated in these surveys. The results revealed that the proportion of participants who reported never having been tested previously for HIV decreased substantially after HBHTS was implemented, from 25% to 7% after 2 rounds of HBHTS. During these first two rounds, 87% to 90% of survey participants who had never tested previously for HIV subsequently accepted HBHTS, and the yield of new HIV diagnoses among those who accepted was

**Table 3. Reasons for never tested for HIV before and intention to have HIV test in the next 12 month among never testers.**

| | 2014 | 2015 | 2016 | 2017 |
|---|---|---|---|---|
| | n = 743, 25% | n = 363, 12% | n = 373, 7% | n = 313, 7% |
| | N (%) | N (%) | N (%) | N (%) |
| **Reasons for never tested** | | | | |
| **Risk Perceptions** | 244 (33) | 58 (16) | 73 (20) | 67 (21) |
| Not at risk for HIV (1) | 240 | 46 | 51 | 51 |
| Too young, need consent (11) | 5 | 12 | 22 | 17 |
| **Fear** | 69 (9) | 25 (7) | 58 (16) | 37 (12) |
| Afraid to learn HIV positive (2) | 64 | 21 | 48 | 27 |
| Afraid of blood, needle or pain (18,24) | 7 | 4 | 10 | 10 |
| **Indifference** | 144 (19) | 75 (21) | 73 (20) | 52 (17) |
| Do not want/care (12, 22) | 29 | 22 | 26 | 9 |
| Do not know/think (16,19) | 112 | 53 | 49 | 43 |
| **Discrimination (26)** | 25 (3) | 5 (1) | 11 (3) | 8 (3) |
| If HIV+, will lose partner/family friends (3) | 3 | 0 | 1 | 1 |
| If HIV+, will be beaten/hurt by partner (4) | 4 | 2 | 2 | 3 |
| Partner does not want me to test (5) | 10 | 2 | 6 | 3 |
| Family/friends do not want me to test (6) | 7 | 1 | 1 | 1 |
| Wait for partner to test together (17) | 3 | 0 | 1 | 0 |
| **Access/Time** | 195 (26) | 127 (35) | 92 (25) | 86 (27) |
| Live too far from testing site (7) | 34 | 5 | 9 | 1 |
| Cost too much money to test (8) | 5 | 0 | 0 | 1 |
| Did not know where to test for HIV (10) | 53 | 31 | 40 | 30 |
| Lack of time (13) | 51 | 42 | 14 | 22 |
| Lack of access, opportunity (14, 21) | 49 | 50 | 29 | 25 |
| Lack of knowledge/information (23) | 10 | 0 | 1 | 0 |
| **Support** | 87 (12) | 43 (12) | 64 (17) | 70 (22) |
| Health provider never offered test (9) | 79 | 43 | 64 | 70 |
| Need encouragement (25) | 8 | 0 | 0 | 0 |
| **Intend to test for HIV in the next 12 months** | | | | |
| Yes | 661 (89) | 313 (87) | 307 (85) | 269 (87) |
| No | 81 (11) | 47 (13) | 53 (15) | 41 (13) |

Participants were allowed to choose more than one reason.

high (11%-13%). Additionally, home rather than a hospital in Chókwè became the most frequently reported most recent HIV testing location. These results suggest that two rounds of HBHTS in high prevalence settings were helpful in substantially reducing the proportion of NTs while achieving a high yield of new HIV diagnoses among NTs tested.

Similar to the findings from a one-time HBHTS campaign [24], repeated HBHTS was well-received by different socio-demographic groups including those who reported various reasons for not having tested for HIV previously. During the four survey rounds, 87%–90% of NTs accepted HBHTS, including close to 90% of adolescents and young NTs (ages 15–24 years), male and female NTs, and NTs with limited access or time for testing, perceived low risk, and lack- of- support for testing. A majority (71%) of NTs who reported fear of learning they were HIV-positive or fear of blood, needles, or pain accepted HBHTS; among those, 9% were diagnosed as HIV-positive.

**Table 4. HBHTC acceptance and HBHTC positive among persons who have not tested before.**

| | Never had HIV test before | Accept HBHTS | HIV positive by HBHTS |
|---|---|---|---|
| | N = 1776 | N = 1570 (88%) | N = 154 (10%) |
| **Time** | | | |
| 2014 | 743 | 668 (90) | 85 (13) |
| 2015 | 360 | 312(87) | 36 (11) |
| 2016 | 365 | 320 (88) | 28 (9) |
| 2017 | 308 | 270 (88) | 5 (2) |
| **Sex** | | | |
| Male | 862 | 752 (87) | 64 (9) |
| Female | 914 | 818 (89) | 86 (11) |
| **Age** | | | |
| $<18$ | 839 | 768 (92) | 11 (1) |
| 18–24 | 336 | 304 (90) | 28 (9) |
| $\geq 25$ | 601 | 498 (83) | 111 (23) |
| **Ever have sex** | | | |
| Yes | 1128 | 985 (87) | 139 (14) |
| No | 642 | 580 (90) | 10 (2) |
| **Never asked partner about HIV status when have sex** | | | |
| Yes | 744 | 657 (88) | 87 (13) |
| No | 936 | 828 (88) | 51 (6) |
| **Reasons for not test** | | | |
| **Not at risk** | | | |
| Yes | 437 | 389 (89) | 26 (7) |
| No | 1339 | 1181 (88) | 124 (10) |
| **Fear** | | | |
| Yes | 185 | 131 (71) | 12 (9) |
| No | 1591 | 1439 (90) | 138 (10) |
| **Indifference** | | | |
| Yes | 340 | 291 (86) | 28 (10) |
| No | 1436 | 1279 (90) | 122 (10) |
| **discrimination** | | | |
| Yes | 48 | 42 (88) | 5 (12) |
| No | 1728 | 1528 (88) | 145 (10) |
| **Lack of access/Time** | | | |
| Yes | 498 | 470 (94) | 54 (11) |
| No | 1278 | 1100 (86) | 96 (9) |
| **Lack of Support** | | | |
| Yes | 264 | 237 (90) | 25 (11) |
| No | 1512 | 1333 (88) | 125 (9) |

There were 14 never testers who participated in HPS for 2 years and 1 who participated for 3 yrs. Only the first observation is included in this analysis.

HIV-positive yield among NTs decreased from 13% during 2014 to 2% during 2017. The decrease in positive yield coincided with a decreasing proportion of NTs who had ever had sex (from 80% during 2014 to 65%, 57%, and 33% in subsequent years) and an increasing proportion of NTs who were aged $<18$ (from 30% during 2014 to 46%, 58%, and 80% in subsequent years). Among NTs aged $\geq 25$ years (97% of whom had ever had sex), the positive yield was ~20% during the first 3 years, decreasing to 5% during 2017. Considering that $<2\%$ of

participants who self-reported never having had sex and accepted HIV testing tested positive, these findings suggest that when HBHTS is repeated in limited-resource settings, screening for having ever had sex might increase HIV-positive yield and reduce the cost associated with testing persons who are pre-sexual debut and are likely not to have not been exposed to HIV.

Strategies for increasing HIV testing uptake among adolescents are important for HIV epidemic control because 14% of all new HIV infections occur during adolescence (ages 10–19 years) [28]. Additionally, during 2005–2012, HIV-related deaths among adolescents increased by 50%, whereas the total number of HIV-related deaths decreased by 30% globally [28]. After adjusting for sexual behaviors, adolescents aged 15–17 remained substantially more likely to have never tested during all 4 years of the CHDSS. Although the yield of new HIV diagnoses among these adolescents was low (1%) and many of them had never had sex (55%), increasing HIV testing uptake for sexually active adolescents can potentially increase early HIV diagnosis and use of ART and thus reduce HIV-related death. The majority of HIV-infected adolescents are unaware of their HIV status and thus haven't initiated ART [28].

Previous findings also indicate that HIV testing might increase HIV knowledge and lead to reductions in sexual risk even when test results are negative [29]. Chókwè is in the province (Gaza) with the highest adult (ages 15–49 years) HIV infection rates in Mozambique [10, 25]; therefore, helping HIV-negative adolescents remain negative by connecting them to youth-friendly prevention services (e.g., family planning, HIV/sexually transmitted infection prevention, and preexposure prophylaxis) and by reinforcing prevention behaviors is particularly important. As part of HBHTS, counselors routinely provided risk reduction counseling, distributed condoms, and provided important information about family planning and treatment for sexually transmitted infections, and circumcision referral for adolescent males and adult men, as warranted.

Men were more likely to be NTs than women during all four survey rounds. This is concerning because 74% of Mozambican girls have their first sexual experience before age 18 and young Mozambican girls commonly have sex with older men, who are potentially at higher risk for HIV infection [30]. Men are also 2–3 times more likely to transmit HIV to women than women are to men [31], yet economic reasons contribute to men's reluctance to access HTS. This might explain why HIV disproportionally affects adolescent girls; 75% of infected adolescents (ages 15–19) in Mozambique are female [30]. Although our results reveal that the HBHTS approach is well-accepted by adolescent NTs and adult male NTs, men have been less easily contactable at home [32, 33]. Additional strategies that specifically target girls and older men to reduce sexual risk and to reach and increase the uptake of HIV testing remain important for HIV epidemic control.

Increased comprehensive knowledge of HIV, including how HIV is transmitted, prevented, and treated, was determined to be negatively associated with NTs (ages 15–59) during all 4 years, after adjusting for sex, age, and sexual behaviors. In contrast, findings in Nigeria and Ghana indicated that HIV testing uptake is low among university students who generally had good knowledge about HIV/AIDS and knew where to get HIV testing [34, 35]. The unwillingness of students to take an HIV test might be attributed to fear, anxiety, and stigma or discrimination. Similarly, from our results, we determined that the fear factor was hardest to overcome; only 71% of NTs who reported fear as a reason for never having been tested accepted HBHTS. This demonstrates that knowing where to test for HIV might be insufficient. Comprehensive knowledge about HIV/AIDS is also important for decreasing fear and stigma or discrimination against PLHIV [36–38] and increasing HIV testing uptake and linkage to care. Increased effort to expand awareness of HIV testing and knowledge is needed, particularly because of the high illiteracy rates in Mozambique (50% among Portuguese speakers and 94% among non-Portuguese speakers) [21].

Affordability, feasibility, and sustainability are key for the success of implementing health programs. Costs-per-person tested and counselled and costs-per-person tested HIV positive are important cost measures for policy makers to decide if the programs can reach the objectives of testing (i.e., increasing HIV testing coverage, identifying PLHIV who were not aware of their status and linking the newly diagnosed PLHIV to treatment). Hauck (2019) reviewed the cost of HBHTS studies in sub-Saharan Africa and concluded that the average cost per person tested for HBHTS was $23 (range: $6 to $55); the average cost per person tested HIV-positive was $439 (range: $66 to $800) [39]. HBHTS was found to be less costly and more effective than FBHTS in increasing HIV testing uptake in rural South Africa [40]. Identifying an optimal frequency of HBHTS has also been discussed. HIV positive yield in the first round depends on the level of prevalence and the testing coverage by existing HIV testing modalities (e.g., FBHTS). The yield of subsequent rounds depends on the HIV incidence and the testing uptake of the prior rounds and other existing modalities. The costs-per-person HIV positive would increase unless HIV uptake among PLHIV were higher in the subsequent rounds [39]. Modeling studies has been done to estimate the optimal frequency of HBHTS [41–43]. Diminishing returns have been suggested but none of the studies offer firm recommendations [39].

For the sensitivity analysis, the estimates using the weighted approach and the observed data were very similar because of random sampling. This confirms that the distributions of race, gender and region of the 10% or 20% random samples of CHDSS households with members aged 15–59 years were similar to the underlying population. When comparing the demographic distributions (e.g., age, gender) of the survey respondents to baseline census data, there were a slightly higher proportion (3–5 percentage points) of older persons (ages 45–59) and a lower proportion (1–10 percentage points) of males among survey respondents. This might be because men are more likely to be away from home during the day and older persons are more likely to be at home. Although the weighted results were similar to the results using observed data, given that 40%-48% of eligible participants were not reached at home or refused to respond during the 4 survey rounds, and we do not know if these not-reached or refused-to-respond eligible participants can be represented by the HPS respondents, we do not generalize the weighted results to all CHDSS participants in the district.

There are several additional limitations of this study. Although HBHTS tested many residents in 2014–2017, including persons who had never tested previously for HIV, other HTS delivery strategies such as routine HIV testing at district health facilities, and school-based and outreach testing for youth and adults, were also being implemented and likely contributed to the reduction of NTs. Additionally, data collected through HPS interviews are subject to recall or reporting bias, particularly for sensitive questions such as drug use and sexual behaviors. Furthermore, the generalizability of the findings might be limited to similar settings as the CHDSS and might not be generalizable to the entire population of Mozambique or Sub-Saharan Africa. However, our results remain informative because a strong similarity exists in the barriers and facilitators to HIV testing across Sub-Saharan Africa, despite the heterogeneity of that region [44].

Future studies are needed for linking those testing HIV-positive with treatment and investigating the need for additional interventions to reach NTs who cannot be reached by HBHTS even after multiple repetitions. Although HBHTS costs have been studied in Kenya and Uganda and HBHTS was reported to compare favorably with other HTS delivery strategies at the time [45, 46], additional cost-effectiveness analyses that compare HBHTS with other delivery strategies (e.g., index testing) and that evaluate the efficiency of repeated HBHTS among populations with different magnitudes of underlying HIV prevalence will be helpful. Despite these limitations, to our knowledge this is the first study to investigate repeated HBHTS in

reaching, testing, and diagnosing HIV infection among NTs and fills a crucial gap in the literature.

## Conclusions

The repeated HBHTS approach was helpful to increase HIV testing coverage and identifying PLHIV in Chókwè. HBHTS acceptance rates were high across all sex, gender and NTs with different barriers. HIV-positive yields among NTs who accepted HBHTS exceeded 10%. In high HIV-prevalence settings with low testing coverage, repeated HBHTS can be considered to increase testing uptake and HIV diagnosis among NTs.

**Disclaimer:** The findings and conclusions in this report are those of the authors and do not necessarily represent the official position of the funding agencies.

## Supporting information

**S1 File. Composite variables.**
(DOCX)

**S2 File. Survey questions.**
(DOCX)

**S1 Table. Demographic and behavioral factors associated with never tested for HIV before by weighted approach.**
(DOCX)

**S2 Table. Reasons for never tested for HIV before and intention to have HIV test in the next 12 month among never testers (by weighted approach).**
(DOCX)

**S3 Table. HBHTC acceptance and HBHTC positive among persons who have not tested before (by weighted approach).**
(DOCX)

**S1 Data. Round1r.**
(XLS)

**S2 Data. Round2r.**
(XLS)

**S3 Data. Round3r.**
(XLS)

**S4 Data. Round4r.**
(XLS)

## Acknowledgments

The authors would like to thank Kay Smith for her helpful editorial review to enhance the manuscript. The authors also would like to acknowledge Dr. Alfredo Vergara, Dawud Ujamaa and Judite Cardoso for their contribution in data acquisition.

## Author Contributions

**Conceptualization:** Carol Lin.

**Data curation:** Isabelle Casavant, Alicia Jaramillo.

**Formal analysis:** Carol Lin, Timothy Green.

**Writing – original draft:** Carol Lin, Timothy Green.

**Writing – review & editing:** Carol Lin, Isabelle Casavant, Alicia Jaramillo, Timothy Green.

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
