## [Decision Letter · Decision Letter 0]

11 Aug 2020

PONE-D-20-18756

Using repeated home-based HIV testing services to reach and diagnose HIV infection among persons who have never tested for HIV, Chókwè Health Demographic Surveillance System, Chókwè District, Mozambique, 2014–2017

PLOS ONE

Dear Dr. Lin:

Thank you for submitting your manuscript to PLOS ONE. After careful consideration, we feel that it has merit but does not fully meet PLOS ONE’s publication criteria as it currently stands. Therefore, we invite you to submit a revised version of the manuscript that addresses the points raised during the review process.

We look forward to receiving your revised manuscript.

Kind regards,

Joseph K.B. Matovu, Ph.D.

Academic Editor

PLOS ONE

Journal Requirements:

2. Thank you for indicating in your methods section that consent was obtained from participants. Please provide additional details regarding participant consent. In your methods section and ethics statement in the online submission form, please ensure that you have specified:

(1) whether consent was informed

(2) what type you obtained (for instance, written or verbal, and if verbal, how it was documented and witnessed).

(3) As your study included people under 18 years old, state whether you obtained consent from parents or guardians, or explain if local law allows for people ages 15 and above to give informed consent for this type of research.

(4) If the need for consent was waived by the ethics committee, please include this information.

"The project has been supported by the President’s Emergency Plan for AIDS Relief (PEPFAR) through the Centers for Disease Control and Prevention (CDC) under the terms of [CoAg#GH00080]."

Reviewers' comments:

Reviewer's Responses to Questions

**Comments to the Author**

1. Is the manuscript technically sound, and do the data support the conclusions?

Reviewer #1: Partly

Reviewer #2: Yes

2. Has the statistical analysis been performed appropriately and rigorously? 

Reviewer #1: Yes

Reviewer #2: Yes

3. Have the authors made all data underlying the findings in their manuscript fully available?

Reviewer #1: No

Reviewer #2: Yes

4. Is the manuscript presented in an intelligible fashion and written in standard English?

Reviewer #1: Yes

Reviewer #2: Yes

5. Review Comments to the Author

Reviewer #1: It is a well written manuscript. The language is simple and easy to understand for readers. However, i believe that the manuscript could be technically more sound especially the methods section. Some steps in the methods section could be explained more vividly to give a better picture of how the study was conducted. Overall, the paper was a good attempt at explaining the HBHTS process.

Reviewer #2: The authors report a study titled “Using repeated home-based HIV testing services to reach and diagnose HIV infection among persons who have never tested for HIV, Chókwè Health Demographic Surveillance System, Chókwè District, Mozambique, 2014–2017”, a serial cross-sectional survey.

The authors cite a high background HIV prevalence in this setting and yet a large proportion ~ 40% of the PLWHIV have never accessed HIV testing and are therefore unaware that they are HIV-infected and not on readily available life-prolonging ART. This study sought to better understand how to bridge the access/utilization gap through a repeat home-based HIV-testing strategy.

With the benefit of survey data from four annual cycles, the research team was able to demonstrate that the highest risk categories of ‘never-tested’ in this study population were adolescents (both boys and girls) and adult men, relative to adult women. Reassuringly, the proportion of never-tested individuals decreased over-time in this relatively stable population; the majority ~ 90% accepted HIV testing and the HIV-positive yield decreased with time. The team concludes that the repeat home -based HIV-testing strategy improved HIV testing in Chókwè study area and recommend implementing similar HIV-testing strategies in high HIV-prevalence settings with low testing coverage, to increase HIV testing uptake and diagnosis among never-tested individuals.

This study presents crucial data to better understand effective HIV testing strategies to increase uptake, especially among those persons who have never been tested. In addition to the information value, the manuscript is well written, and therefore should be considered for publication. There were minor observations for consideration/clarification.

Methods:

1. Lines 111-114 need revision for grammatical typos..

Statistical analysis.

1. Lines 143-145. “The observed data were analyzed using SAS® (version 9.3, SAS Institute, Inc., Cary, North Carolina, USA) survey procedures that account for correlations among participants within a household following by a sensitivity analysis.”

a. It is prudent to indicate the survey method(s) used to account for correlations with citation where applicable.

b. Sentence needs revision re: grammar.

2. Lines 147-151. Please indicate the statistical method(s) used for the two sample comparisons of the proportions.

Results:

1. Lines 203-211. It would be interesting to know how many of those who self-reported never having had sex and accepted testing were HIV-positive?

2. Lines 212-213 need revision. Not clear.

Discussion:

1. Lines 242-249. The authors state in their discussion “These findings suggest that when HBHTS is repeated, screening for sexual risk behavior should be considered to avoid unnecessary costs associated with testing persons who are pre-sexual debut and are likely not to have not been exposed to HIV.”

This was overstated and not entirely supported by the data as presented. While, the authors write, “Among NTs aged ≥25 years (97% of whom had sex), the positive yield was ~20% …”, there was no mention of the HIV-positive yield among the 3% self-reported virgins? What about the adolescents who self-reported never had sex?

Additionally, self-reported sexual behavior variable has limitations (desirability bias), prone to misclassification of the intended construct. Caution should be exercised in making strong recommendations. Also, please note that there have been occurrences of young adolescents with no sexual experience who were unaware of their perinatally acquired HIV-infection. Lastly, the authors articulate in another section of the discussion about the potential behavior change benefits after a negative test, so it would be counterproductive to deny an HIV-screen test in this high-risk age group under the pretext of a self-reported sexual virginity.

6. PLOS authors have the option to publish the peer review history of their article (what does this mean?). If published, this will include your full peer review and any attached files.

Reviewer #1: No

Reviewer #2: No

---

## [Author Response · Author response to Decision Letter 0]

1 Sep 2020

Please see the attached file, response to reviewers, for the point-to-point response to the editor and reviewers. Thank you!!

---

## [Decision Letter · Decision Letter 1]

2 Oct 2020

PONE-D-20-18756R1

Using repeated home-based HIV testing services to reach and diagnose HIV infection among persons who have never tested for HIV, Chókwè Health Demographic Surveillance System, Chókwè District, Mozambique, 2014–2017

PLOS ONE

Dear Dr. Carol Lin:

Thank you for submitting your manuscript to PLOS ONE. After careful consideration, we feel that it has merit but does not fully meet PLOS ONE’s publication criteria as it currently stands. Therefore, we invite you to submit a revised version of the manuscript that addresses the points raised during the review process.

This paper is well written and I would like to commend the authors for this. Here are a few minor comments that the authors need to address prior to acceptance:

1. Repeat HBHTS is a useful and powerful approach to reach non-testers. However, home-based HIV testing comes at a cost and most programs, including governments -- especially in LMICs -- might not be able to implement HBHTS without additional donor support. I suggest that the authors include a discussion around the cost of HBHTS vis-a-vis the scalability and sustainability of such programs in poor-resource settings.

2. The authors report that the HIV yield reduced from 13% in 2014 to 2% in 2015-17. Similar reductions were observed across sex and age-groups. I expected to see a clear discussion on why HIV prevalence declined significantly in the four-year period. Definitely, I would imagine that HBHTS is not the reason for the decline; so, what caused the decline? Even if the authors did not set out to explore why the HIV yield declined, it would be important to discuss what could have caused the observed decline in the HIV yield over time. Is it that HBHTS reached less risky individuals over subsequent rounds? Characteristically speaking, would the authors argue that NTs were generally a high-risk group (initially, e.g. in 2014) but that the demographic profile changed with continued repeat HBHTS?

3. Related to item 2 above; as the HIV yield reduces significantly, the cost per HIV-positive person identified increases subsequently. I think this aspect should be discussed along with the authors' response to item 1 above.

4. In line 243, the authors should use 'were' in place of 'was' after '...high prevalence settings...'

5. In lines 256-57, reference is made to 6% during 2017 but in lines 218-19, this percentage is given as 5%. The authors should harmonize these figures.

We look forward to receiving your revised manuscript.

Kind regards,

Joseph K.B. Matovu, Ph.D.

Academic Editor

PLOS ONE

Reviewers' comments:

Reviewer's Responses to Questions

**Comments to the Author**

1. If the authors have adequately addressed your comments raised in a previous round of review and you feel that this manuscript is now acceptable for publication, you may indicate that here to bypass the “Comments to the Author” section, enter your conflict of interest statement in the “Confidential to Editor” section, and submit your "Accept" recommendation.

Reviewer #2: All comments have been addressed

2. Is the manuscript technically sound, and do the data support the conclusions?

Reviewer #2: Yes

3. Has the statistical analysis been performed appropriately and rigorously? 

Reviewer #2: Yes

4. Have the authors made all data underlying the findings in their manuscript fully available?

Reviewer #2: Yes

5. Is the manuscript presented in an intelligible fashion and written in standard English?

Reviewer #2: Yes

6. Review Comments to the Author

Reviewer #2: (No Response)

7. PLOS authors have the option to publish the peer review history of their article (what does this mean?). If published, this will include your full peer review and any attached files.

Reviewer #2: No

---

## [Author Response · Author response to Decision Letter 1]

16 Oct 2020

1.Repeat HBHTS is a useful and powerful approach to reach non-testers. However, home-based HIV testing comes at a cost and most programs, including governments -- especially in LMICs -- might not be able to implement HBHTS without additional donor support. I suggest that the authors include a discussion around the cost of HBHTS vis-a-vis the scalability and sustainability of such programs in poor-resource settings.

Response: Thanks for the recommendation. We have added the following paragraph to the Discussion Section.

“Affordability, feasibility, and sustainability are key for the success of implementing health programs. Costs-per-person tested and counselled and costs-per-person tested HIV positive are important cost measures for policy makers to decide if the programs can reach the objectives of testing (i.e., increasing HIV testing coverage, identifying PLHIV who were not aware of their status and linking the newly diagnosed PLHIV to treatment). Hauck (2019) reviewed the cost of HBHTS studies in sub-Saharan Africa and concluded that the average cost per person tested for HBHTS was $23 (range: $6 to $55); the average cost per person tested HIV-positive was $439 (range: $66 to $800) [39]. HBHTS was found to be less costly and more effective than FBHTS in increasing HIV testing uptake in rural South Africa [40]. Identifying an optimal frequency of HBHTS has also been discussed. HIV positive yield in the first round depends on the level of prevalence and the testing coverage by existing HIV testing modalities (e.g., FBHTS). The yield of subsequent rounds depends on the HIV incidence and the testing uptake of the prior rounds and other existing modalities. The costs-per-person HIV positive would increase unless HIV uptake among PLHIV were higher in the subsequent rounds [39]. Modeling studies has been done to estimate the optimal frequency of HBHTS [41-43]. Diminishing returns have been suggested but none of the studies offer firm recommendations [39].”

2. The authors report that the HIV yield reduced from 13% in 2014 to 2% in 2015-17. Similar reductions were observed across sex and age-groups. I expected to see a clear discussion on why HIV prevalence declined significantly in the four-year period. Definitely, I would imagine that HBHTS is not the reason for the decline; so, what caused the decline? Even if the authors did not set out to explore why the HIV yield declined, it would be important to discuss what could have caused the observed decline in the HIV yield over time. Is it that HBHTS reached less risky individuals over subsequent rounds? Characteristically speaking, would the authors argue that NTs were generally a high-risk group (initially, e.g. in 2014) but that the demographic profile changed with continued repeat HBHTS?

Response: The homebased HIV testing program was part of HIV combination prevention programs and other HTS delivery strategies such as routine HIV testing at district health facilities; school-based and outreach testing were also being implemented. The population profile that HBHTS reached was similar overtime. 

However, the population profile of never testers did change over time: there were more younger people (age<18) and more participants who reported never having had sex. On the other hand, the estimated prevalence overall in Chokwe ranged from 28% to 25% during the study period. So, yes, HBHTS reached less risky individuals over subsequent rounds. As we discussed in the Discussion Section: 

 “HIV-positive yield among NTs decreased from 13% during 2014 to 2% during 2017. The decrease in positive yield coincided with a decreasing proportion of NTs who had ever had sex (from 80% during 2014 to 65%, 57%, and 33% in subsequent years) and an increasing proportion of NTs who were aged <18 (from 29% during 2014 to 46%, 58%, and 79% in subsequent years). Among NTs aged ≥25 years (97% of whom had sex), the positive yield was ~20% during the first 3 years, decreasing to 6% during 2017. Considering that <2% of participants who self-reported never having had sex and accepted HIV testing tested positive, these findings suggest that when HBHTS is repeated in limited-resource settings, screening for having ever had sex might increase HIV-positive yield and reduce the cost associated with testing persons who are pre-sexual debut and are likely not to have not been exposed to HIV.”

3. Related to item 2 above; as the HIV yield reduces significantly, the cost per HIV-positive person identified increases subsequently. I think this aspect should be discussed along with the authors' response to item 1 above.

Response:This is a good point. Please see our addition to the discussion in our response to item 1.

4. In line 243, the authors should use 'were' in place of 'was' after '...high prevalence settings...'

Response: Corrected. Thank you!

5. In lines 256-57, reference is made to 6% during 2017 but in lines 218-19, this percentage is given as 5%. The authors should harmonize these figures.

Response: Corrected. Thank you!

---

## [Editor Report · Decision Letter 2]

19 Oct 2020

PONE-D-20-18756R2

Using repeated home-based HIV testing services to reach and diagnose HIV infection among persons who have never tested for HIV, Chókwè Health Demographic Surveillance System, Chókwè District, Mozambique, 2014–2017

PLOS ONE

Dear Dr. Carol Lin,

Thank you for submitting your manuscript to PLOS ONE. After careful consideration, we feel that it has merit but does not fully meet PLOS ONE’s publication criteria as it currently stands. Therefore, we invite you to submit a revised version of the manuscript that addresses the points raised during the review process.

A rebuttal letter that responds to each point raised by the academic editor. You should upload this letter as a separate file labeled 'Response to Reviewers'.A marked-up copy of your manuscript that highlights changes made to the original version. You should upload this as a separate file labeled 'Revised Manuscript with Track Changes'.An unmarked version of your revised paper without tracked changes. You should upload this as a separate file labeled 'Manuscript'.

We look forward to receiving your revised manuscript.

Kind regards,

Joseph K.B. Matovu, Ph.D.

Academic Editor

PLOS ONE

Additional Editor Comments (if provided):

I would like to thank the authors for adequately responding to my comments.

As we move to the finishing line, the authors should now pay attention to the following errors in the 'References' section:

1. PLoS ONE uses the Vancouver style while formatting references. The authors should check refs 9,27, 39 and 43 and ensure these are well aligned to the journal's referencing style. The authors may find it helpful to visit this link for additional information: https://journals.plos.org/plosone/s/submission-guidelines#loc-references 

2. In ref 40, the journal name is not provided.

3. The authors should ensure that journal names are written in a consistent way, for example, in ref 4, the journal name is given as "PLoS One"; while in ref 8, the same journal is given as "Plos one". The authors should check other journal names to correct them accordingly.
---

## [Author Response · Author response to Decision Letter 2]

19 Oct 2020

1. PLoS ONE uses the Vancouver style while formatting references. The authors should check refs 9,27, 39 and 43 and ensure these are well aligned to the journal's referencing style. The authors may find it helpful to visit this link for additional information: https://journals.plos.org/plosone/s/submission-guidelines#loc-references

Response: The references are checked and updated to be aligned to the journal’s referencing style. Please see the updated manuscript. 

2. In ref 40, the journal name is not provided.

Response: The journal name is added. Thank you!!

3. The authors should ensure that journal names are written in a consistent way, for example, in ref 4, the journal name is given as "PLoS One"; while in ref 8, the same journal is given as "Plos one". The authors should check other journal names to correct them accordingly.

Response: The journal names are checked and updated. Thank you!!

---

## [Editor Report · Decision Letter 3]

30 Oct 2020

Using repeated home-based HIV testing services to reach and diagnose HIV infection among persons who have never tested for HIV, Chókwè Health Demographic Surveillance System, Chókwè District, Mozambique, 2014–2017

PONE-D-20-18756R3

Dear Dr. Carol Lin:

We’re pleased to inform you that your manuscript has been judged scientifically suitable for publication and will be formally accepted for publication once it meets all outstanding technical requirements.

Kind regards,

Joseph K.B. Matovu, Ph.D.

Academic Editor

PLOS ONE
---

## [Editor Report · Acceptance letter]

11 Nov 2020

PONE-D-20-18756R3 

Using repeated home-based HIV testing services to reach and diagnose HIV infection among persons who have never tested for HIV, Chókwè Health Demographic Surveillance System, Chókwè District, Mozambique, 2014–2017 

Dear Dr. Lin:

I'm pleased to inform you that your manuscript has been deemed suitable for publication in PLOS ONE. Congratulations! Your manuscript is now with our production department. 

Kind regards, 

on behalf of

Dr. Joseph K.B. Matovu 

Academic Editor

PLOS ONE